# Single-Cell Atlas of Patient-Derived Trophoblast Organoids in Ongoing Pregnancies

Olivier J. M. Schäffers [1,2], Catherine Dupont [2], Eric M. Bindels [3], Diane Van Opstal [4], Dick H. W. Dekkers [5], Jeroen A. A. Demmers [5], Joost Gribnau [2,*] and Bas B. van Rijn [1,*]

1    Department of Obstetrics and Gynaecology, Division of Obstetrics and Fetal Medicine, Erasmus University Medical Center, 3015 CN Rotterdam, The Netherlands
2    Department of Developmental Biology, Oncode Institute, Erasmus University Medical Center, 3015 GD Rotterdam, The Netherlands
3    Department of Hematology, Erasmus University Medical Center, 3015 GD Rotterdam, The Netherlands
4    Department of Clinical Genetics, Erasmus University Medical Center, 3015 CN Rotterdam, The Netherlands
5    Proteomics Center, Erasmus University Medical Center, 3015 GD Rotterdam, The Netherlands
*    Correspondence: j.gribnau@erasmusmc.nl (J.G.); b.vanrijn@erasmusmc.nl (B.B.v.R.)

**Abstract:** Trophoblast organoids (TOs) hold great promise for elucidating human placental development and function. By deriving TOs in ongoing pregnancies using chorionic villus sampling (CVS), we established a platform to study trophoblast differentiation and function in early pregnancy, including pregnancies with different fetal genetic abnormalities. We addressed cellular heterogeneity of CVS-derived TOs by providing a single-cell transcriptomic atlas and showed that CVS-TOs recapitulate key aspects of the human placenta, including syncytial fusion and hormone synthesis. This study demonstrates the utility of trophoblast organoids for investigating genetic defects in the placenta and describes an experimental platform for future personalized placental medicine approaches, including genotype–phenotype mapping.

**Keywords:** placenta; trophoblast organoids; chorionic villus sampling; Down syndrome; Cornelia de Lange syndrome

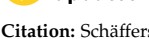



## 1. Introduction

The placenta is a significant determinant of fetal development and lifelong health [1]. Placental abnormalities have been associated with pregnancy disorders, including miscarriage, preterm birth, preeclampsia and fetal growth restriction [2]. In humans, however, placental development and function are notoriously difficult to study due to the obvious challenges in obtaining and sustaining placental tissue during ongoing pregnancy [1]. Trophoblast organoids (TOs) have recently been derived from first-trimester tissue and hold promise to model key features of placental development and function [3,4]. These include the maintenance of cytotrophoblast (CTB) stemness and differentiation of these cells into syncytiotrophoblast (SCT), a multinucleated layer responsible for nutrient exchange and hormone production, and extravillous cytotrophoblast (EVT), invasive cells that anchor the placenta to the uterine wall and remodel maternal spiral arteries [5,6].

Currently, TO models have only been derived from early pregnancy terminations, limiting their translational potential as clinical tools during pregnancy, including genotype-phenotype mapping and drug screening [7]. To advance TO models towards patient-derived platforms that allow for personalized interrogation of placental development and function, we established a method for the derivation of TOs from chorionic villi obtained in ongoing pregnancies ranging from 10 to 14 weeks of gestation. Furthermore, we employed single-cell transcriptional profiling to interrogate cellular heterogeneity of TOs and highlight trophoblast transition states during early pregnancy. Our work confirms that TOs mimic key aspects of the human placenta and provides a new platform to investigate the impact of genetic abnormalities on placental development and function.

## 2. Materials and Methods

### 2.1. Tissue Collection

First-trimester placental tissues (10–14 weeks of gestation) were obtained from chorionic villus sampling (CVS) procedures. CVS was routinely performed at Erasmus University Medical Center for prenatal diagnosis in pregnancies at increased genetic risk. Utilization of tissues and all experimental procedures were performed according the regulatory framework provided by the Erasmus University Medical Center Ethics Board (study number: OZBS71.19172) and are in accordance with the guidelines in The Declaration of Helsinki 2000.

### 2.2. Trophoblast Organoid Formation and Culture

To enrich for trophoblast suspensions, 5–10 mg of CVS tissue was enzymatically digested with 0.25% trypsin-EDTA (Gibco, Waltham, MA, USA) for 30 min in a humidified incubator at 37 °C and 5% $CO_2$. Cell suspensions were filtered through a 40-μm cell strainer (Corning, Tewksbury, MA, USA), washed in Advanced DMEM/F12 medium (Gibco, Waltham, MA, USA) and centrifuged at $300\times g$ for 5 min. Cells were plated in GFR Matrigel (Corning) and cultured in trophoblast organoid medium (TOM) consisting of Advanced DMEM/F12 supplemented with 1X B27 minus vitamin A (Life Technologies, Carlsbad, CA, USA), 1X N2 (Life Technologies), 100 μg/mL primocin (Invivogen, San Diego, CA, USA), 2 mM L-glutamine (Sigma, Saint Louis, MO, USA), 1.25 mM N-Acetyl-L-cysteine (Sigma), 500 nM A83-01 (Tocris, Bristol, UK), 1.5 μM CHIR99021 (Tocris), 50 ng/mL recombinant human EGF (Gibco), 80 ng/mL recombinant human R-spondin1 (Peprotech, London, UK), 100 ng/mL recombinant human FGF2 (Peprotech), 50 ng/mL recombinant human HGF (Gibco), 2 μM Y–27632 (Merck Millipore, Burlington, MA, USA), and 2.5 μM prostaglandin E2 (Sigma), as previously described [4]. Medium was replaced every 2–3 days and organoids were passaged every 7 to 10 days depending on their size and density. For passaging, organoid-containing drops of Matrigel were mechanically dissociated by extensive pipetting (200–500 times) using small pipette tips until organoids were broken into hardly visible particles. Organoids were washed with ice-cold Advanced DMEM/F12 to remove any remaining Matrigel. For induction of EVT differentiation, organoids were cultured in TOM for 7–10 days and switched to TOM without CHIR99021 and R-spondin1. Outgrowth of cells was observed after 3–5 days. Organoids were cryopreserved in Cellbanker-2 (Amsbio, Abingdon, UK) and stored in liquid nitrogen.

### 2.3. Immunofluorescence and Confocal Microscopy

Whole-mount immunofluorescence staining of trophoblast organoids was performed as previously described [8]. The following primary antibodies were used: monoclonal anti-CDH1 (rabbit, 1:200, Cell Signaling, 24E10), monoclonal anti-KRT7 (mouse, 1:100, DAKO, M7018) and polyclonal anti-GDF15 (rabbit, 1:100, Sigma, HPA011191). Before imaging, organoids were embedded in Matrigel drops in the center of 60 mm Petri dishes. Imaging was performed using the Leica SP5 Intravital confocal microscope with a HCX APO 20x water dipping objective. Images were processed with ImageJ.

### 2.4. Single-Cell Isolation for 10X Genomics

Roughly 50 day-14 trophoblast organoids were pooled and dissociated into single cells. For this, organoid-containing drops of Matrigel were dissolved in ice-cold Cell Recovery solution (Corning) for 30 min at 4 °C. Next, organoids were collected and incubated with 0.25% trypsin-EDTA (Gibco) for 15 min at 37 °C, followed by extensive pipetting (100–300 times) using small pipette tips. Cells were filtered through a 40-μm cell strainer (Falcon), centrifuged at $300\times g$ for 5 min and checked for viability. Finally, cells were resuspended in Advanced DMEM/F12 medium (Gibco) to a final concentration of 700–1000 cells/μL.

### 2.5. Library Preparation and Sequencing

Single cells were processed on the 10× Genomics Chromium Platform using the Chromium Next GEM Single Cell 3′ Reagent kit v3.1 with Dual Index Kit (10× Genomics) following the manufacturer's protocol. Briefly, 15,000 cells were loaded in each channel of a chip to be partitioned into gel beads in emulsion (GEMs). Within GEMs, cells were lysed followed by barcoded reverse transcription of RNA. Breaking of the GEMs was followed by amplification, fragmentation and addition of adapter and sample index. Libraries were pooled and sequenced on an Illumina NovaSeq 6000 instrument (minimum coverage of 25,000 raw reads per cell).

### 2.6. Single-Cell Data Analysis

Raw FASTQ files were quantified and aligned to the GRCh38 human reference genome using the Cell Ranger software pipeline (version 4.0.0, 10× Genomics, Pleasanton, CA, USA). Cells with fewer than 500 counts, 250 detected genes and low complexity (log10 value of number of genes detected per count < 0.8) were removed. Genes with zero expression in all cells and that are expressed in less than 10 cells were removed. Downstream data analyses were performed using the R package Seurat (version 4.0.3, Satija lab, New York, NY, USA) [9]. Datasets were normalized by the 'SCTransform' function. Cell cycle heterogeneity was scored and differences between S and G2M cell cycle phases were regressed out using the functions 'CellCycleScoring' and 'ScaleData'. Datasets were integrated by canonical correlation analysis following the Seurat alignment workflow. UMAP analysis was performed using the 'RunUMAP' function including 20 principal components. Clusters were identified using the 'FindClusters' function with a resolution of 0.6. Cluster marker genes were identified by differential expression analysis using the 'FindAllMarkers' function with a minimum log fold change value > 0.25, adjusted $p$-value < 0.01 and genes detected in a minimum of 25% cells in each cluster. Clusters were annotated using canonical trophoblast markers and gene signatures of existing single-cell datasets of first-trimester placental tissue [10,11] were plotted using the 'AddModuleScore' function. Trajectory modeling and pseudotemporal ordering of cells was performed with Monocle 3 and ArchR packages [12]. For Monocle 3, cells were clustered using the 'find_clusters' function with UMAP as reduction method, the principal graph was created with the 'learn_graph' function and cells were ordered using the 'order_cells' function with CTB progenitors as root node. For ArchR, diffusion plots were created with the Density package and cells were aligned to the trajectory based on their Euclidean distance with calculated pseudotime values based on these distances for a number of iterations using the functions 'addTrajectory' and 'plotTrajectory'. Senescence signatures were obtained from the Human Ageing Genomic Resources database [13] and Reactome Pathway (ID: R-HAS-2559582.2). Permutation test to calculate relative proportional cluster differences were performed with the R library 'scProportionTest' available at https://gitbhub.com/rpolicastro/scProportionTest/ (accessed on 1 December 2021).

### 2.7. Genotyping

Genomic DNA was extracted from trophoblast organoids using the PicoPure™ DNA extraction kit (Thermo Fisher Scientific, Waltham, MA, USA) following the manufacturer's instructions. DNA quality and concentration were determined on a Nanodrop ND-1000 Spectrophotometer.

Genotyping of DS-TOs was performed by single nucleotide polymorphism (SNP) array analysis. Per sample, 50 to 100 ng of genomic DNA was hybridized to the Illumina Infinium GSA+MD-24 v3 BeadChip genotyping array. Data analysis for copy number calls was performed at genome-wide resolution of 5 Mb using Genome Studio (Illumina, San Diego, CA, USA) and NxClinical (BioDiscovery, Ann Arbor, MI, USA) software. Genotyping of CdLS-TOs was performed by Sanger sequencing. The *SMC1A* target region (exon 4-5) was amplified by PCR and send for Sanger sequencing.

*2.8. Mass Spectrometry*

Organoid supernatant was collected at day 14 of culture, heat inactivated for 10 min at 80 °C and 1000 rpm and then reduced in volume using a Speedvac (Thermo Fisher Scientific). Samples were diluted with MilliQ water and 1M Tris-HCl (pH 7.4) to a final concentration of 200 mM Tris-HCl and SDS powder (Genomics Solutions Inc., Ann Arbor, MI, USA) was added to a final concentration of 1%. Samples were lysed by heating for 30 min at 60 °C and 1000 rpm. Proteins were reduced by addition of 1/20th volume of 100 mM DTT (Sigma) and incubated for 45 min at 50 °C and 1000 rpm. Next, samples were cooled to room temperature and reduced proteins were alkylated by addition of 1/20th volume of 200 mM freshly prepared 2-cloro-acetamide (18.7 mg/mL, Fluka) and incubated in the dark for 45 min at room temperature and 1000 rpm. The alkylation process was quenched by addition of another 1/20th volume of 100 mM DTT and incubation for 15 min at room temperature and 1000 rpm. Samples were centrifuged at 12,500× *g* for 10 min and clear supernatants were transferred to clean tubes. Samples were processed further by the SP3 method as previously described [14] using a bead concentration of 0.5 μg/μL and 1 M salt concentrations. Peptides were then analyzed by nanoflow liquid chromatography tandem mass spectrometry (nLC-MS/MS), which was performed on an EASY-nLC coupled to an Orbitrap Fusion Tribrid mass spectrometer (Thermo Fisher Scientific) operating in positive mode. Peptides were separated on a ReproSil-C18 reversed-phase column (Dr Maisch; 15 cm × 50 μm) using a linear gradient of 0–80% acetonitrile (in 0.1% formic acid) during 90 min at a rate of 200 nL/min and corresponding elutions were sprayed directly into the electrospray ionization (ESI) source of the mass spectrometer. Spectra were acquired in continuum mode and fragmentation of the peptides was performed in data-dependent mode by HCD. Raw mass spectrometry data were analyzed using the MaxQuant software suite [15]; version 2.0.1.0, Max Planck Institute, Martinsried, Germany) as described previously [16] with the additional options 'LFQ' and 'iBAQ' selected. The following settings were applied: peptide-spectrum match and protein false discovery rate of 0.01; minimum peptide length of 7 amino acids; a maximum of two missed cleavages; peptide tolerance of 10 ppm and fragment ion tolerance of 0.6 Da for HCD spectra; enzyme specificity was set to trypsin and cysteine; carbamidomethylation was set as a fixed modification. The Andromeda search engine was used to search the MS/MS spectra against the Uniprot database (taxonomy: *Homo sapiens* and *Mus musculus*, release June 2021) concatenated with the reversed versions of all sequences. In case the identified peptides of two proteins were the same or the identified peptides of one protein included all peptides of another protein, these proteins were combined by MaxQuant and reported as one protein group. Before further statistical analysis, known contaminants and reverse hits were removed.

## 3. Results

First, we developed and tested a method to establish TOs from ongoing pregnancies using chorionic villus sampling (CVS) tissue (Figure 1a). For this, surplus chorionic villi that were sampled for prenatal diagnosis between 10 to 14 weeks of gestation were used. We dissociated villous protrusions from the mesenchymal core to enrich for CTB populations (Supplementary Figure S1a). To assess the heterogeneity of the isolated suspensions, we performed single-cell transcriptional profiling of 6481 cells. Isolated suspensions consisted mainly of CTBs (89% of all cells; *KRT7*$^+$, *CDH1*$^+$, *VIM*$^-$) (Supplementary Figure S1b,c). We seeded the isolated suspensions as single cells into Matrigel drops that we cultured in previously defined trophoblast organoid medium [4]. Homogeneous organoid cultures established after 11–14 days of culture (Figure 1b). All cultures showed similar morphology and growth rates. Whole mount immunohistochemical staining showed that CVS-derived TOs (CVS-TOs) consist of mononuclear CTBs expressing *CDH1* and *KRT7* that line cavities filled with multinucleated cells expressing SCT marker *GDF15* (Figure 1c). This indicates that CTBs spontaneously fuse into SCT towards the center of the organoid structures. As previously described [3,4], omission of Wnt stimulators from the organoid cultures resulted in the outgrowth of EVT-like cell populations (Supplementary Figure S1d). In-depth

characterization and functional analysis of EVT populations was not performed. Overall, we could derive TOs from a variety of CVS tissues with an efficiency of 72% (18 out of 25 samples, Supplementary Table S1) and up to 14 weeks of gestation, the start of the second trimester. Previous TO models have only been reported from first trimester placentas up to 9 weeks of gestation. Cultures of CVS-TOs could be established for at least 2 months, cryopreserved and re-cultured (data not shown).

Second, we used single-cell transcriptional profiling to study trophoblast differentiation in CVS-TOs at day 14 of culture. Dissociation and pooling of organoids (*n* = 3 samples) for single-cell analysis resulted in transcriptomic information for a total of 15,642 cells after quality control. Graph-based clustering resolved seven trophoblast clusters (Figure 1d). We plotted gene signatures from available single-cell atlases of early placental tissues [10,11] to pinpoint CTB and SCT populations (Figure 1e). We identified four CTB states *(KRT7$^+$, PAGE4$^+$, PEG10$^+$)* and three SCT states *(GDF15$^+$, SDC1$^+$, KISS1$^+$)* based on differential gene expression (Figure 1d–f, Supplementary Table S2). Our data are consistent with the hypothesis that SCTs are replenished from a population of CTB progenitor cells [17]. In CVS-TOs, CTB progenitor cells *(ITGA6$^+$, EPCAM$^+$, MKI67$^+$)* express a range of integrins and members of Wnt and EGF signaling pathways to maintain their stemness and proliferation, while genes implicated in syncytial fusion and hormone synthesis are upregulated towards SCT end-point states (Figure 1e,f).

Next, we validated the ability of CVS-TOs to model key trophoblast cell state changes during placental development using trajectory analysis with ArchR (Figure 1g). More specifically, we validated the behavior of known regulators of trophoblast fate *TEAD4*, *OVOL1* and *MSX2*. In line with previous reports [18–20], CTB progenitor cells in CVS-TOs lose their self-renewal capacities in the transition to a differentiated CTB state, marked by *TEAD4* downregulation and transient *OVOL1* upregulation, that subsequently requires *MSX2* downregulation for activation of the SCT transcriptional program (Figure 1g).

We then assessed whether CVS-TOs could be derived from pregnancies with fetal genetic abnormalities, such as chromosomal aneuploidies and single gene mutations. With our method, we could successfully derive TOs from pregnancies where the fetus was affected with Down syndrome (DS-TOs; trisomy 21) and Cornelia de Lange syndrome (CdLS-TOs; *SMC1A* mutation; c.532A>C) (*n* = 1 each; Supplementary Figure S2a,b). Both syndromes are associated with poor pregnancy outcomes and placental abnormalities [21,22]. We integrated single-cell transcriptional profiles of DS-TOs (*n* = 12,816) and CdLS-TOs (*n* = 15,509) with control CVS-TOs at day 14 of culture and obtained 44,226 cells after quality control (Figure 2a). CTB and SCT states were largely conserved between all conditions (Supplementary Figure S3a, Supplementary Table S2), with an additional population of progenitor CTBs expressing stress makers such as *FOS* and *JUN* mostly present in CdLS-TOs (Figure 2a, Supplementary Figure S4). Strikingly, the majority of cells within CdLS-TOs showed signs of defective cell cycling (negative G2M and S scores) suggestive of growth arrest (Figure 2b). Cellular senescence is associated with growth arrest and has been reported in placentas of CdLS mouse models [22]. We plotted senescence-associated gene signatures and indeed mostly observed genes upregulated in senescence in (terminally) differentiated SCT populations (Figure 2c). In contrast, CdLS-TOs showed increased expression of senescence-related transcripts in CTB states, with high levels of transcripts related to the senescence-associated secretory phenotype. We further hypothesized that senescence in early trophoblast differentiation (e.g., CTB state) in CdLS-TOs might associate with decreased SCT formation, and performed a permutation test to calculate the relative differences in cluster proportions between all CVS-TO conditions. Indeed, CdLS-TOs were depleted for all SCT states compared with DS-TOs and control CVS-TOs (FDR < 0.05 and mean log2 fold enrichment > 1; Supplementary Figure S3b). Conversely, CdLS-TOs were enriched for early CTB states. To further validate SCT depletion in CdLS-TOs, we performed an unbiased proteomic analysis of organoid supernatants at day 14 (*n* = 4) to detect SCT-specific proteins. We found lower amounts of SCT-secreted human placental lactogen (CSH1) and pregnancy-specific glycoprotein 8 (PSG8) in supernatant of CdLS-TOs compared to control CVS-TOs (Figure 2d). We

observed no relevant differences in cluster proportions and SCT-secreted proteins between DS-TOs and control CVS-TOs (Figure 2d, Supplementary Figure S3c).

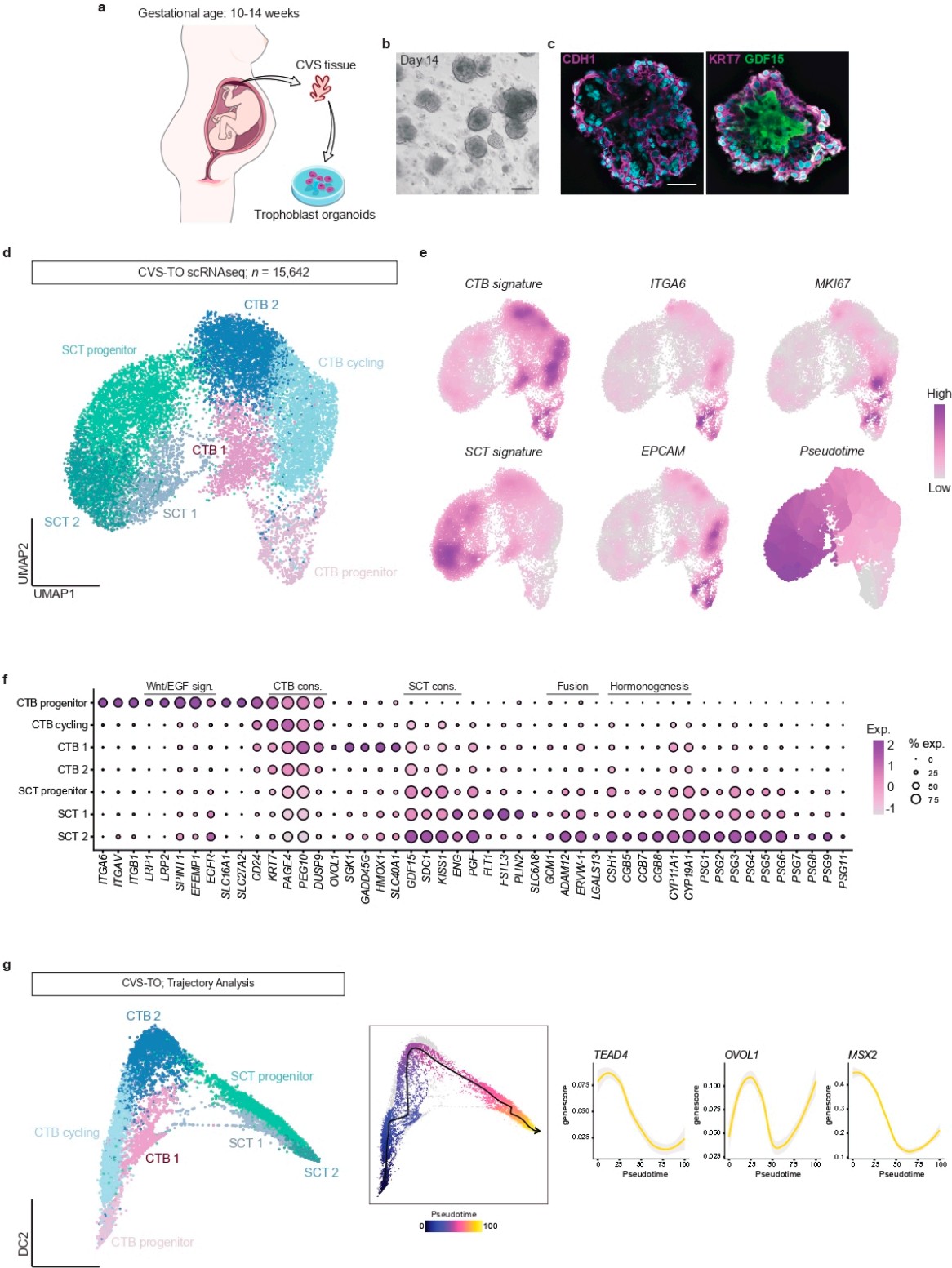

**Figure 1.** Derivation and single-cell transcriptional profiling of CVS-TOs. (**a**) Schematic overview illustrating the derivation of TOs from CVS tissue. (**b**) Brightfield image of CVS-TOs at day 14. Scale bar, 200 μm. (**c**) Representative whole mount staining of CVS-TOs at day 14 showing expression of

*CDH1, KRT7* and *GDF15*. Nuclei are stained with Hoechst. Scale bar, 100 μm. (**d–g**) Single-cell profiling of pooled CVS-TOs (*n* = 3 samples) at day 14. (**d**) Uniform Manifold Approximation and Projection for Dimension Reduction (UMAP) plot showing 7 distinct trophoblast clusters (*n* = 15,642; SCT progenitor, *n* = 3866; CTB cycling, *n* = 3337; CTB 2, *n* = 3283; SCT 2, *n* = 1984; CTB 1, *n* = 1284; SCT 1, *n* = 1043; CTB progenitor, *n* = 845). (**e**) UMAP plots showing normalized expression of CTB and SCT signatures obtained from first-trimester single-cell atlases, progenitor markers *ITGA6, EPCAM* and *MKI67* and cells colored by relative Monocle3 pseudotime value. (**f**) Dotplot showing normalized expression and percentage of expressing cells for a range of trophoblast markers. (**g**) Diffusion plot showing the trajectory analysis of CVS-TO at day 14. Pseudotime values were plotted using ArchR. Gene scores (approximation for expression) were plotted against pseudotime for known regulators of SCT fate *TEAD4, OVOL1* and *MSX2*.

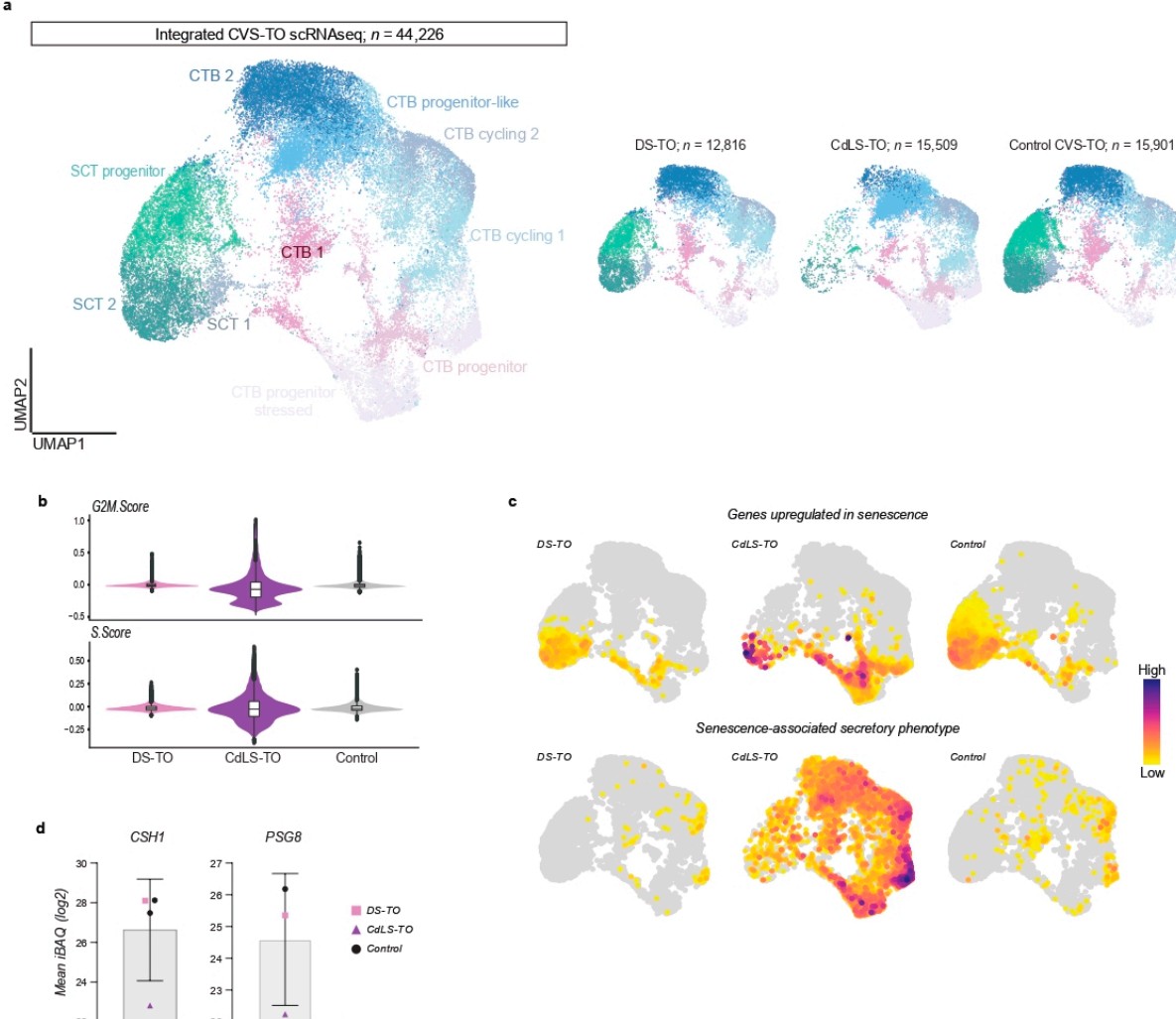

**Figure 2.** Derivation of CVS-TOs from pregnancies with genetic abnormalities. (**a–c**) Single-cell profiling of DS-TOs (*n* = 1 sample) and CdLS-TOs (*n* = 1 sample) at day 14. (**a**) UMAP plot showing 10 distinct trophoblast clusters after integration of single-cell transcriptional profiles of DS-TOs (*n* = 12,816), CdLS-TOs (*n* = 15,509) and control CVS-TOs (*n* = 15,901) at day 14 of culture. (**b**) Violin plot showing G2M and S cell cycle scores. (**c**) UMAP plots showing normalized expression of senescence signatures obtained from the Human Ageing Genomic Resources database and Reactome pathways. (**d**) Abundance of CSH1 and PSG9 proteins in CVS-TO culture supernatants (*n* = 4 and *n* = 3, respectively) at day 14. Results are expressed in mean log2 iBAQ values.

## 4. Discussion

In summary, we have shown that TOs derived from CVS tissue combined with single-cell transcriptional profiling can be used to capture fundamental states in placental development and function. To date, TO models have only been derived from terminated pregnancies up to 9 weeks of gestation and single-cell transcriptomic atlases of TOs are limited to studies describing trophoblast stem cell-derived TOs [3,4,23,24]. Here, we provide transcriptional data of TOs that span into the early second trimester, a time critical to placental development and differentiation. Our findings further highlight the potential of CVS-TOs to capture individual disease phenotypes; modeling placental-specific effects of genetic abnormalities in respect to their native genetic background, including those underlying developmental disorders such as Cornelia de Lange syndrome.

We note that this study has some limitations. Due to the relatively low incidence of Down syndrome and Cornelia de Lange syndrome our experiments were conducted in a relatively small cohort. Notwithstanding, we still think that the reported data provides an interesting resource for future studies. Considering the anonymized nature of the tissue collection protocol extensive clinical information was not available during this study. Questions for future investigation will be to integrate TO data with clinical and genetic parameters, as has been described for other organoid models [7,25], further exploring the potential of TOs as predictors of pregnancy success and childbirth. Additionally, investigation of EVT lineage development was limited. In-depth characterization and functional analysis of EVT populations would strengthen the application of CVS-TOs, providing a platform to study EVT dynamics in early placental disorders or pregnancies with fetal genetic abnormalities.

In conclusion, making use of surplus CVS tissue derived in ongoing pregnancies, our method could be readily aligned with standard prenatal diagnostic pipelines, provides a valuable step forward to the implementation of TO models for better understanding of placental development and function, and opens up avenues towards patient-derived trophoblast organoids as a platform for disease modeling, drug testing and personalized therapy.

**Supplementary Materials:** The following supporting information can be downloaded at: https://www.mdpi.com/article/10.3390/organoids1020009/s1, Figure S1: Derivation of CVS-TOs; Figure S2: Genotyping of CVS-TOs; Figure S3: Clustering and cluster proportions of integrated single-cell transcriptional profiles of DS-TOs, CdLS-TOs and control CVS-TOs; Figure S4: Population of progenitor CTBs expressing stress markers; Table S1: Summary of CVS-TO derivation and characterization; Table S2: Differential gene expression analysis in CVS-TOs; Table S3: LC-MS/MS analysis of CVS-TO supernatants.

**Author Contributions:** Conceptualization, O.J.M.S. and B.B.v.R.; methodology, O.J.M.S., D.V.O. and B.B.v.R.; validation, O.J.M.S.; formal analysis, O.J.M.S.; investigation, O.J.M.S., E.M.B., D.H.W.D. and J.A.A.D.; writing—original draft preparation, O.J.M.S.; writing—review and editing, O.J.M.S., C.D., J.G. and B.B.v.R.; visualization, O.J.M.S.; supervision, C.D., J.G. and B.B.v.R.; funding acquisition, B.B.v.R. All authors have read and agreed to the published version of the manuscript.

**Funding:** This work was supported by personal grant to Bas van Rijn from the Erasmus MC Sophia Foundation (funding number: CAM20-13).

**Institutional Review Board Statement:** Utilization of tissues and all experimental procedures were performed according the regulatory framework provided by the Erasmus University Medical Center Ethics Board (study number: OZBS71.19172) and are in accordance with the guidelines in The Declaration of Helsinki 2000.

**Informed Consent Statement:** Patient consent was waived in line with the existing Dutch ethical regulations for the use of surplus human tissue, and within the appropriate regulatory framework for anonymized use of surplus human tissue provided by the Erasmus University Medical Center Ethics Board.

**Data Availability Statement:** All data associated with this study are present in the paper or the Supplementary Information. Raw sequencing data have been deposited in the NCBI GEO repository (GSE211429) and will be publicly available as of the date of publication. Code used for analysis is available upon request.

**Acknowledgments:** We acknowledge support from the Department of Clinical Genetics with collection of the CVS tissue. We thank Jonathan Windster for fruitful discussions.

**Conflicts of Interest:** The authors declare no competing interest.

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
