# Peer review of "Single-Cell Atlas of Patient-Derived Trophoblast Organoids in Ongoing Pregnancies"

_2674-1172, doi:10.3390/organoids1020009_

Round 1

Reviewer 1 Report

In this manuscript Schaffers et al. present an exciting application of CVS tissue specimens for the generation of trophoblast organoids. The topic of this research study is timely and advances the field through the application of a unique and underutilized tissue source, CVS, to identify how early events of placentation and genetic abnormalities may be linked with pregnancy outcomes. This article is well organized and clearly written. A few points outlined below will help to improve the clarity and impact of the findings presented.

1.              The authors mentioned associating clinical data with TO outcomes as a future direction. Some general info regarding the range of gestational ages is provided. Can any sample metadata for the tissues used in this study be provided at a single sample level (i.e., gestational age at collection, amount of tissue, fetal sex, etc.)? Similarly, could the authors include information on clinical genotyping for purposes of comparison to the TO genotyping following culture?

2.              The number of cells analyzed is clearly reported. However, it is unclear how many CVS tissue specimens (control and otherwise) are used to generate the single cell datasets. Were the DS and CdLS specimens n=1 each? If so, are the findings reported representative?

3.              The authors show limited data regarding EVT cell lineage development using the TO model (phase contrast images in Supp Fig 2). Further characterization of the EVT cells (transcriptomic, ICC, etc.) would strengthen the proposed application of this model to include both terminal trophoblast lineages. Alternatively, acknowledgment of the limitation regarding studying EVT cell lineage development should be included in the discussion. Was EVT cell development assessed in DS-TO or CdLS-TOs?

4.              The DS-TOs seemed to largely be used as another control population for comparing to the CdLS-TOs. Were any unique characteristics of the DS-TO observed?

Author Response

The authors would like to thank the reviewer for the positive notes and provided suggestions. Careful consideration was given to the reviewer’s suggestions after which we tried to adjust the manuscript accordingly. Please find every point addressed in attached file.  

Reviewer 2 Report

In this manuscript, Schäffers et al. describe the derivation of trophoblast organoids from chorionic villus biopsies of pregnancies at risk for genetic diseases. They describe this methodology for apparently genetically normal samples obtained from 10-14 weeks of gestation, which exceeds the capability of stem/progenitor cell derivation from previous publications by several weeks. They also manage to derive TOs from CVSs that were genotyped as Trisomy 21 (Down syndrome) and Cornelia de Lange Syndrome (SMC1A mutation). Of all these TOs, the authors present single cell RNA-sequencing data and interpretations thereof, with the main conclusion being that CdLS-TOs exhibit signs of premature senescence even in cytotrophoblast cells.

Overall, this is a very valid study that expands previous horizons on trophoblast organoids considerably.

Several points should be addressed to improve the manuscript:

1. Critically, information must be provided about the number of CVS samples in each category. From Figure 2D it would appear that there are only 1 trisomy 21 and 1 CdLS sample each, and 2 controls, that were analyzed. Is this really the case? It conflicts with the information that 18/25 TO derivation attempts were successful. Also, why are there 2 controls in the CSH1 plot but only 1 control data point in the PSG8 plot in Fig. 2D? Please provide details about patient numbers per group in the text and in a Table, listing how many of them resulted in successful TOs, and what TOs of which patient samples were further analyzed.

2. In addition to this information, details should be provided about the heterogeneity within each group for parameters such as TO growth rates, morphological/histological differences etc.

3. Was the single cell analysis performed on TOs from individual patients or only on pooled TOs per group (i.e. 3 in total)? Please spell out this information in the text, and add it to the table requested in Point 1 if appropriate.

4. From the results described, it could be expected that CdLS TOs proliferate less and may fail to self-renew in culture over time. Was this the case?

5. The authors derive TOs from 10-14 wk pregnancies. This was attempted before but failed. Did the authors improve culture conditions for trophoblast cells from these later gestational stages to maintain their proliferative capacity?

6. Patient consent information must be provided.

7. Complete results of the mass spectrometry analysis should be provided.

8. The Supplementary Figures should be improved, in particular figures S1-S3. They could be combined. Please make an effort to make clear what is shown.

Author Response

(The authors gave the same response as above.)
